Allocation pattern and accumulation potential of carbon stock in natural spruce forests in northwest China

Yue Jun-Wei 1 2 3
Guan Jin-Hong 2
Deng Lei 4
Zhang Jian-Guo 5
Li Guoqing 1 2
Du Sheng 1 2 shengdu@ms.iswc.ac.cn
1 State Key Laboratory of Soil Erosion and Dryland Farming on Loess Plateau, Institute of Soil and Water Conservation, Chinese Academy of Sciences and Ministry of Water Resources , Yangling, Shaanxi , People’s Republic of China
2 Institute of Soil and Water Conservation, Northwest A&F University , Yangling, Shaanxi , People’s Republic of China
3 University of Chinese Academy of Sciences , Beijing , People’s Republic of China
4 Academy of Agriculture and Forestry Sciences, Qinghai University , Xining, Qinghai , People’s Republic of China
5 Upper and Middle Yellow River Bureau, Yellow River Conservancy Commission of the Ministry of Water Resources , Xi’an, Shaanxi , People’s Republic of China
Marino Bruno
Electronic publication date: 2018 May 25
Publication date: 2018
Volume: 6
Electronic Location ID: e4859
Received 2018 Jan 30; Accepted 2018 May 8
Copyright: © 2018 Yue et al.
Copyright year: 2018
Copyright holder: Yue et al.
License: This is an open access article distributed under the terms of the Creative Commons Attribution License, which permits unrestricted use, distribution, reproduction and adaptation in any medium and for any purpose provided that it is properly attributed. For attribution, the original author(s), title, publication source (PeerJ) and either DOI or URL of the article must be cited.
License URL: https://creativecommons.org/licenses/by/4.0/

Keywords: Spruce forests, Stand age, Carbon stock, Carbon sequestration potential

Funding: Strategic Priority Research Project of the Chinese Academy of Sciences XDA05050202 The study was supported by the Strategic Priority Research Project of the Chinese Academy of Sciences (XDA05050202): Current stocks, sequestration rate and potential of forest carbon in the temperate western China. The funders had no role in study design, data collection and analysis, decision to publish, or preparation of the manuscript.

==============================
Background

The spruce forests are dominant communities in northwest China, and play a key role in national carbon budgets. However, the patterns of carbon stock distribution and accumulation potential across stand ages are poorly documented.

Methods

We investigated the carbon stocks in biomass and soil in the natural spruce forests in the region by surveys on 39 plots. Biomass of tree components were estimated using allometric equations previously established based on tree height and diameter at breast height, while biomass in understory (shrub and herb) and forest floor were determined by total harvesting method. Fine root biomass was estimated by soil coring technique. Carbon stocks in various biomass components and soil (0–100 cm) were estimated by analyzing the carbon content of each component.

Results

The results showed that carbon stock in these forest ecosystems can be as high as 510.1 t ha−1, with an average of 449.4 t ha−1. Carbon stock ranged from 28.1 to 93.9 t ha−1 and from 0.6 to 8.7 t ha−1 with stand ages in trees and deadwoods, respectively. The proportion of shrubs, herbs, fine roots, litter and deadwoods ranged from 0.1% to 1% of the total ecosystem carbon, and was age-independent. Fine roots and deadwood which contribute to about 2% of the biomass carbon should be attached considerable weight in the investigation of natural forests. Soil carbon stock did not show a changing trend with stand age, ranging from 254.2 to 420.0 t ha−1 with an average of 358.7 t ha−1. The average value of carbon sequestration potential for these forests was estimated as 29.4 t ha−1, with the lower aged ones being the dominant contributor. The maximum carbon sequestration rate was 2.47 t ha−1 year−1 appearing in the growth stage of 37–56 years.

Conclusion

The carbon stock in biomass was the major contributor to the increment of carbon stock in ecosystems. Stand age is not a good predictor of soil carbon stocks and accurate evaluation of the soil carbon dynamics thus requires long-term monitoring in situ. The results not only revealed carbon stock status and dynamics in these natural forests but were helpful to understand the role of Natural Forest Protection project in forest carbon sequestration as well.

Introduction

The spruce forests are important communities in dark coniferous forests of the north temperate zone, and are a major forest type in northwestern China. Most of the forests are on high mountains with vertical distribution ranging elevations from 1,200 to 3,800 m. According to the 7th national forest inventory of China in 2008, spruce forests covered an area of approximately 3.94 × 106 ha and their stem volume was 9.9 × 108 m3 in the country (Chinese Ministry of Forestry, 2010). As spruce forests generally have large carbon densities in their vegetation biomass and soil layers (Wagner et al., 2015; Zhou, Yu & Zhao, 2000), there should be substantial amount of carbon storage and sequestration potential in these ecosystems. In the context of climate change, spruce forests are particularly important when assessing the carbon balance dynamics of high mountain forests.

Globally, a lot of work has been carried out to study allocation patterns and accumulation potential of carbon stock in forest ecosystems in the past few decades (Brown & Lugo, 1984; Dixon et al., 1994; Fang et al., 2001; Liu et al., 2014). However, considerable problems and uncertainties still exist in such studies. First, different methodologies may result in errors of carbon stock estimation. For example, as an important approach, the application of model simulation is limited by the assumption and applicable scope when estimating carbon stock in different ecosystems and the estimation error is difficult to measure (Piao et al., 2009; Preece et al., 2015; Quinkenstein & Jochheim, 2016). The method using a variable of Biomass Expansion Factor, another commonly applied method for estimating forest biomass, also has great error when up-scaling the carbon stock estimation from sample plot to a region (Fang et al., 2001; Sun et al., 2016). Secondly, carbon stock of tree layer in forests is commonly measured (Chen & Luo, 2015; Wang et al., 2014), but other components, such as understory vegetation (shrubs, herbs), litter, deadwoods and fine roots have seldom been calculated because of costing much time and labor, although these components may contribute to considerable amounts of carbon stock and the cycling of forest ecosystems (Sun et al., 2015; Takahashi et al., 2010). Fine roots and deadwoods are active participants of carbon cycling and may play important roles in soil organic carbon accumulation. Fine roots are chiefly responsible for water and nutrient uptake into plants (Sun et al., 2015), and also are important sources of carbon in litter inputs within forest soils (Joslin et al., 2006; Leppalammi-Kujansuu et al., 2014). Deadwoods act as a carbon and nutrient reservoir (Chambers et al., 2000), and it could increase soil organic carbon sequestration in forest topsoil (Wambsganss, Stutz & Lang, 2017). Finally, compared with the analyzed carbon content, the use of generalized coefficients (0.5 or 0.45) for estimating carbon stock would lead to obvious error (Bhadwal & Singh, 2002; Fang et al., 2001; Lewis et al., 2009). Therefore, estimation based on comprehensive and exact surveys is needed to improve our understanding of carbon stock and sequestration potential in forest ecosystems.

Forest development has a close relationship with carbon stock over the entire life cycle of ecosystems, as tree growth rates vary greatly with stand age (Law et al., 2003; Taylor, Wang & Chen, 2007). Stand age is one of the critical factors in regulating carbon stock and the partitioning among different components such as the trees, litter and soil (Fonseca, Rey Benayas & Alice, 2011; Law et al., 2003; Martin et al., 2005; Seedre et al., 2015). To our knowledge, however, most studies focused mainly on the dynamics of carbon stock in tree and soil while the carbon stock in understory (shrub, herb), litter, fine root and deadwood have seldom been considered. Meanwhile, the change of carbon accumulation along with stand age in soil (increase or stable) has been a controversial issue (Peichl & Arain, 2006; Zhao et al., 2014). It is thus necessary to explore the long-term changes of carbon accumulation and allocation in forest ecosystems.

Picea asperata and Picea crassifolia are dominant conifer species and are also unique tree species in northwest China (Gou et al., 2005). This study took the two species as model to quantify the carbon accumulation and allocation among the various biomass components (tree, understory, litter, deadwood, fine root) and soil across an age sequence in northwest China. The overall goals were to: (1) estimate the carbon stock and allocation pattern of all ecosystem components; (2) assess the carbon accumulation potential. The information from this study is expected to provide more accurate evaluation of the carbon sequestration potential in dark coniferous forests, and also is crucial to evaluate spruce forest ecosystem services.

Materials and Methods

Site description

The study area (coverage coordinates 35°40′–38°44′N and 99°49′–106°17′E) is located in Gansu and Ningxia provinces in northwest China (Fig. 1). From 1981 to 2010, annually mean temperature ranged from 0.3 to 10.0 °C and mean annual precipitation ranged from 171.1 to 492 mm (National Meteorological Information Center, 2012). Most of the rainfall occurs from July to September and the period of the highest temperature spans from June to August. Spruce forest is one of the major forest types in the two provinces and was protected after the Natural Forest Protection project (NFP) was launched in 1998. Picea asperata and Picea crassifolia are dominant species, Populus davidiana, Salix caprea, Salix rehderiana, and B. utilis are the common companion species in tree layer, Elaeagnus pungens, Syringa oblata, Spiraea pubescens, Rosa multiflora, Potentilla fruticosa are dominant species in shrub layer, and Carex tristachya is the dominant species in herbaceous layer. The typical soil types are dark-brown earth, meadow sierozem and cinnamon soil (National Soil Survey Office, 1998).

Figure 1 Location of the study area and sampling plots.

Dot points represent the geographical location of 39 sample plots of spruce forest.

Sample plots establishment and sampling

The sampling plan was implemented according to the method of scaling down: grid-site-plot, following the IPCC (2003). The specific method of establishing sample plots followed the guidance of Observation and Investigation for Carbon Sequestration in Terrestrial Ecosystems (Technical Manual Writing Group of Ecosystem Carbon Sequestration Project, 2015). The number and location of sample plots of major communities were determined according to the weight of area, volume, distribution range, age class, which were referred from previous national forest surveys. Meanwhile, the forests with different age classes should be incorporated in the sample plots, so that the sample plots have the greatest representative significance for the forest type. According to the principles mentioned above, totally 39 representative sampling plots with the dimensions 20 m × 50 m (several plots were slightly smaller depending on topography limitation) were established in spruce forests. These sampling plots are free from human interference and are distributed at an elevation range between 2,300 and 3,200 m a.s.l. Within each sampling plot, three representative shrub subplots with the dimensions 2 m × 2 m were established along a diagonal of the sampling plot, while three subplots with dimensions 1 m × 1 m were set up along the other diagonal for sampling of herbage and litter. The sample survey was conducted between July and September 2012.

For each plot, tree height (H) and diameter at breast height (DBH) were measured for all individuals with DBH ≥2 cm. Meanwhile, latitude, longitude, elevation, slope position, gradient and species names were recorded. Stand age for each plot was estimated by coring three biggest trees inside the plot. The average ring count of the tree samples is used as the stand age of each plot (Van Tuyl et al., 2005). Stand age of all sample plots ranged from 34 to 141 years and was divided into four age classes. The basic characteristics for the sample plots are summarized in Table 1.

Table 1 Site characteristics of the natural spruce forests.

Stand age classes	I (<40 year)	II (40–60 year)	III (60–80 year)	IV (>80 year)	
No. of plots	6	20	9	4	
Elevation (m)	2,548 ± 86	2,801 ± 34	2,715 ± 59	2,849 ± 101	
Density (trees ha−1)	2,877 ± 305	1,682 ± 165	1,591 ± 344	1,637 ± 148	
Height (m)	5.2 ± 0.1	11.0 ± 0.1	9.3 ± 0.2	8.8 ± 0.2	
DBH (cm)	7.9 ± 0.1	14.4 ± 0.2	15.1 ± 0.4	16.4 ± 0.5	
Basal area (m2 ha−1)	19.0 ± 1.7	38.2 ± 3.5	39.7 ± 3.9	49.7 ± 5.1	

For each dominant tree species, three to five sample trees with a small, medium and large DBH were selected for sampling to determine carbon content of tree components. Samples of stem, branch, needle and root (about 300 g) from each sample tree were collected and brought back to the laboratory for determining carbon content.

In each shrub and herb subplot, all aboveground organs were harvested, and the roots were excavated from the soil. The aboveground components of shrubs were further divided into leaves and branches. The fresh weight of each component was weighed in situ and subsamples were collected for determining carbon content and estimating dry biomass. Litter was collected and weighed in the field within the herbage subplots. Samples were also collected and brought back to laboratory for determining moisture and carbon content.

Downed dead trees were defined as those lying or leaning (with a zenith angle ≤45°) with mid-length diameter ≥2 cm and length ≥1 m. Mid-length diameters and length of all downed dead trees were measured in each plot. Biomass was calculated using measured mid-length diameters, length and decomposition class specific density. Downed dead trees were divided into three grades according to the degree of decay: (1) mild decay, recently dead, some branches and foliage present; (2) moderate decay, there has been some loss of wood; (3) severe decay, there has been more loss of wood, and the wood is very fragile. One disc was collected from each sampled dead wood to determine the density. At least 10 samples were collected from each tree species at each decomposition class (Technical Manual Writing Group of Ecosystem Carbon Sequestration Project, 2015). The same carbon content value was assumed as for the live trees.

Standing dead trees (with a zenith angle ≥45 and height ≥1.3 m) were divided into two categories: one kind was leafless with almost complete crown and looked like live tree, the other kind was incomplete with fractured trunk or crown. Their biomass (including roots) were calculated differently. Biomass in the first kind of standing dead trees was calculated similarly to live trees (using allometric equations). Biomass in the latter kind was estimated by calculating their volume using a truncated cone formula and converted to biomass using the specific deadwood density (Technical Manual Writing Group of Ecosystem Carbon Sequestration Project, 2015). Height of an individual was calculated using the height-diameter curves supplied by Zeng et al. (2009); and dead coarse root biomass was calculated similarly as for the first kind (mild decay) of standing dead trees. The same carbon content values were assumed as for the live trees to estimate the carbon stock of dead trees.

In each plot, 10 representative core-samples were taken for soil down to a depth of 100 cm by the serpentine sampling method. Soil samples representing depths of 0–10, 10–20, 20–30, 30–50 and 50–100 cm were collected using a soil sampling auger (4 cm diameter). Samples from the same depth layer in each plot were mixed for a more representative sample to measure the soil organic carbon content at that depth. In addition, a profile of 100 cm-deep was dug and samples were extracted from five depth (0–10, 10–20, 20–30, 30–50 and 50–100 cm) using ring samplers to measure the soil bulk density.

While coarse roots (diameter ≥2 mm) were included in the calculation by the methods of biomass equations and harvesting, sampling of fine roots was separately conducted. Soil coring technique was used to estimate fine root biomass, including live and dead roots that were less than 2.0 mm in diameter. Samples were randomly taken from ten points in each plot. A 4 cm diameter steel auger was used to sample soil cores of depths in 0–20 and 20–40 cm. Samples were soaked in flowing water, rinsed and sieved out the roots, collected fine roots were reserved for determining biomass and carbon content.

Determination of carbon content and calculation of carbon stock

Biomass of each component (stem, branch, needle and root) for trees was calculated through allometric biomass equations (Table 2) based on DBH and H. Biomass equations were obtained from previous studies (Cheng et al., 2007; Forest Carbon Sequestration Project Office, 2014). Biomass in shrub, herbage and litter layer were determined by total harvesting approach mentioned above.

Table 2 Biomass equations of the dominant species and three common companion species.

Species No.	Stem	Branch	Needle	Root	Bark	
1	e−3.9744(D2H)0.9434	e−4.6350(D2H)0.9257	e−5.9391(D2H)0.9753	e−5.2791(D2H)0.9457	e−5.5587(D2H)0.8930	
2	0.4944(D2H)0.6370	1.0157(D2H)0.4372	1.3368(D2H)0.2537	0.5060(D2H)0.4969		
3	0.021(D2H)0.9642	0.0011(D2H)1.1909	0.0022(D2H)0.8595	0.053(D2H)0.7452		
4	e−3.8023(D2H)0.9631	e−5.9070(D2H)1.0903	e−3.9108(D2H)0.6104	e−3.2756(D2H)0.7692		
Notes:

Species No.: (1) P. asperata and P. crassifolia, (2) Populus davidiana, (3) Salix caprea and S. rehderiana, (4) Betula utilis. Equations for species No. 1 and 4 were obtained from Cheng et al. (2007); and No. 2 and 3 were obtained from the protocol edited by Forest Carbon Sequestration Project Office (2014).

The plant and litter samples were oven dried at 70 °C until constant weight for calculating the biomass. Soil samples for determination of soil organic carbon content were air-dried and ground to pass through 2 mm sieve firstly to remove fine roots and other debris. Then, a part of screened soil was sampled through quartering and further ground until pass through 0.25 mm sieve prior to the laboratory analysis (National Agricultural Technology Extension Service Center, 2006). The carbon content of plant, litter and soil was analyzed using the traditional method of potassium dichromate oxidation-external heating.

The soil organic carbon stock (t ha−1) was calculated using the following equation:(1) SOCS=∑m=15Cm×BDm×Dm×(1−Gm)/10

where Cm is soil organic carbon content (g kg−1) in the soil layer m (total five sampling layers in this study), BDm, Dm and Gm are the bulk density (g cm−3), thickness (cm) and the volumetric fraction of stones (>2 mm) in the corresponding soil layer, respectively.

Forest biomass carbon stock (t ha–1) was calculated as: (2) BCS=∑i=14(CtiBti)+∑j=13(CsjBsj)+∑k=12(ChkBhk)+CdwBdw+CfBf+ClBl

where i represents the tree component (i.e., stem, branches, needles and roots), Cti and Bti are the carbon content and biomass of the corresponding tree component, respectively; j is the shrub component (i.e., branches, leaves and roots), Csj and Bsj are the carbon content and biomass of the shrub component, respectively; k is the component of herbs (i.e., the aboveground and belowground), Chk and Bhk are the carbon content and biomass of the herb component, respectively; the other three items are carbon contents and biomass for deadwood, fine roots and litter, respectively. Carbon stock and biomass follow the same unit of t ha–1, while the carbon content should fall in a range of 0–1. Total carbon stock in the ecosystem was the sum of those in biomass and soil.

Calculation of the carbon accumulation potential and rates of stand biomass

According the description concerning carbon sequestration potential reported by Keith et al. (2010) and Li & Liu (2014), the carbon sequestration potential (t ha−1) of different stand age classes were calculated as: (3) CSPn=CCC−CCSn

where n is the stand age class (n = I, II, II, III and IV), CSPn is carbon sequestration potential in age class n; CCC is carbon carrying capacity of referred ecosystem, which is determined as the mean biomass carbon stock of three plots with the largest values; CCSn is the current biomass carbon stock of age class n, and it is the average value of biomass carbon stocks for all sample plots in the age class.

Carbon sequestration rate was estimated for three stages corresponding to the intervals of four age classes, i.e., I–II, II–III and III–IV. The age intervals were determined by referring to the mean ages for the sample plots in corresponding classes as 21, 14 and 34 years for stages I–II (37–56 years), II–III (56–70 years) and III–IV (70–104 years), respectively. The difference in biomass carbon stocks between two age classes was divided by the age interval to give the carbon sequestration rate of a corresponding stage.

Statistical analysis

The effects of stand age on carbon stock in trees, understory, forest floor, soil, and the total ecosystem were tested using one-way ANOVA. All data were checked for normality and homogeneity of variances prior to one-way ANOVA and the results conformed to the normal distribution. Differences among the means of carbon stocks and carbon contents were determined with the LSD test at a significance level of p < 0.05. Exponential rise to maximum or exponential decay models were used to express the relationships between carbon stock of each component and stand age; the coefficient of determination (R2) and the level of probability (p) were used to determine the goodness of fitting. Statistical analyses were performed using SPSS 20.0 for windows (SPSS 20.0; SPSS Inc., Chicago, IL, USA).

Results

Carbon stocks in trees, shrubs and herbage

Carbon stocks of tree components and the total tree layer increased with stand age (Table 3). Stems contributed the most part to the carbon stock of trees, and the contribution percentage ranged from 52.3% in age class I to 57.8% in age class IV. The average proportion of stem, branch, needle and root relative to the total carbon stock in tree layer were 54.3, 19.6, 10.8 and 15.3%, respectively. The carbon stock in the total tree layer steadily increased along the age sequence, from 28.1 t ha−1 in age class I to 93.9 t ha−1 in age class IV.

Table 3 Distribution pattern and dynamics of carbon stock in tree layer (t ha−1).

Tree component	Stand age classes	
I (<40 yr)	%	II (40–60 yr)	%	III (60–80 yr)	%	IV (>80 yr)	%	Average	%	
Stem	14.7 ± 26a	52.3	42.0 ± 4.5b	52.6	52.3 ± 7.8b	56.5	54.3 ± 3.3b	57.8	41.4 ± 3.5	54.3	
Branch	4.2 ± 0.6a	15.1	17.5 ± 1.9b	21.8	16.6 ± 3.7b	18.0	15.0 ± 4.3ab	16.0	15.0 ± 1.5	19.6	
Needle	4.2 ± 1.2a	14.7	8.3 ± 0.9b	10.4	9.9 ± 1.3b	10.7	9.9 ± 0.8b	10.6	8.2 ± 0.6	10.8	
Root	5.0 ± 1.1a	17.9	12.1 ± 1.2b	15.2	13.7 ± 1.8b	14.8	14.7 ± 0.6b	15.6	11.7 ± 0.9	15.3	
Tree layer	28.1	100	79.9	100	92.4	100	93.9	100	76.3	100	
Note:

For each component, data with different superscript letters are statistically different among different stand age classes. Data are expressed as mean and standard errors. Carbon stock for stem in this study was the sum of stem wood and bark.

Carbon stocks in shrub and herb layers varied from 0.14 to 1.26 t ha−1 and 0.27 to 1.8 t ha−1, respectively, and did not statistically different among the age classes (Table 4). The proportion in branches, leaves and roots within the shrub layer were 41.38%, 6.90% and 51.72%, respectively. In herb layer, aboveground and belowground accounted for 57.89% and 42.11%, respectively. Carbon stocks in shrub and herb layers were independent of stand age (p > 0.05) based on regression analysis (Figs. 2A and 2B).

Table 4 Carbon stock in shrubs and herbs (t ha−1) and the percentage (%) in different stand age classes.

Components	Stand age classes	
I (<40 yr)	%	II (40–60 yr)	%	III (60–80 yr)	%	IV (>80 yr)	%	Average	%	
Branch	0.58 ± 0.31a	65.91	0.07 ± 0.03b	50.00	0.52 ± 0.16a	41.27	0.40 ± 015a	50.77	0.24	41.38	
Leaf	0.02 ± 0.0a	2.27	0.03 ± 0.02a	21.43	0.07 ± 0.02a	5.56	0.05 ± 0.01a	5.87	0.04	6.90	
Root	0.28 ± 0.19a	31.82	0.04 ± 0.02a	28.57	0.67 ± 0.24a	53.17	0.34 ± 0.14a	43.36	0.3	51.72	
Shrub total	0.88	100	0.14	100	1.26	100	0.78	100	0.58	100	
Aboveground	0.11 ± 0.04a	20.37	1.17 ± 0.25b	65.00	0.16 ± 0.03a	32.65	0.14 ± 0.03a	48.15	0.66	57.89	
Belowground	0.43 ± 0.27a	79.63	0.63 ± 0.11a	35.00	0.33 ± 0.10a	67.35	0.14 ± 0.03a	51.85	0.48	42.11	
Herb total	0.54	100	1.80	100	0.49	100	0.27	100	1.14	100	
Note:

For each component, data with different superscript letters are statistically different among different stand age classes. Data are mean and standard errors.

Figure 2 The relationships between carbon stock in different components and stand age.

(A) Shrub layer; (B) herb layer; (C) biomass.

Carbon stocks in soil

Soil organic carbon content decreased significantly with increasing soil depth in all age classes (Table 5). The average soil organic carbon content decreased from 84.3 g kg−1 in 0–10 cm to 30.0 g kg−1 in 50–100 cm. Soil organic carbon contents in 30–50 and 50–100 cm showed statistically difference among age classes, whereas there was no significant variation with stand ages in upper soil layers.

Table 5 Soil organic carbon content at different soil depths with stand age class (g kg−1).

Soil layer (cm)	Stand age class	
I (<40 yr)	II (40–60 yr)	III (60–80 yr)	IV (>80 yr)	Average	
0–10	91.71 ± 13.8Aa	81.74 ± 5.61Aa	86.48 ± 15Aa	81.25 ± 14.12Aa	84.30 ± 4.99	
10–20	64.8 ± 14.49Aab	73.12 ± 4.8Aa	63.95 ± 10.48Aab	48.61 ± 3.02Ab	67.21 ± 4.13	
20–30	46.75 ± 12.12Abc	57.68 ± 4Ab	41.01 ± 9.27Abc	31.36 ± 4.84Abc	49.45 ± 3.7	
30–50	31.38 ± 14.85ABbc	50.69 ± 4.73Bbc	27.93 ± 6.98Ac	24.55 ± 5.18Ac	39.79 ± 3.92	
50–100	15.01 ± 7.12Ac	41.05 ± 5.63Bc	21.17 ± 5.52Ac	17.06 ± 7.67Ac	30.00 ± 3.82	
Notes:

Data with different lowercase letters are significantly different among different soil depths within the same age class, while those with different uppercase letters are significantly different among different age classes within the same horizon (p < 0.05). Data are mean and standard errors.

Soil organic carbon stocks in 50–100 cm soil layers changed significantly with stand ages, but did not show a clear trend (Table 6). Those in other layers did not show significant difference among age classes. The soil organic carbon stored in deep soil (50–100 cm) contributed 26.4%, 45.7%, 39.6% and 25.2% to the total of 0–100 cm in age class I, II, III and IV, respectively, lower than that in the upper 50 cm layer. Soil carbon stock in the 0–100 cm ranged from 254.2 to 420.0 t ha−1 and did not show a changing trend with stand age.

Table 6 Soil organic carbon stock (t ha−1) and percentage (%) indifferent soil layers.

Soil depth (cm)	Stand age classes	
I (<40 yr)	%	II (40–60 yr)	%	III (60–80 yr)	%	IV (>80 yr)	%	Average	%	
0–10	70.4 ± 9.0a	23.1	48.1 ± 4.7a	11.5	46.7 ± 3.8a	15.4	67.4 ± 20.6a	26.5	53.2 ± 3.7	14.9	
10–20	52.7 ± 7.39a	17.2	50.1 ± 5.4a	11.9	47.9 ± 4.1a	15.8	39.8 ± 5.3a	15.7	48.9 ± 3.1	13.6	
20–30	43.2 ± 7.8a	14.1	44.1 ± 3.4a	10.5	33.4 ± 6.6a	11.0	30.9 ± 2.4a	12.2	40.1 ± 2.7	11.2	
30–50	58.9 ± 26.4a	19.2	85.6 ± 8.8a	20.4	55.8 ± 14.3a	18.3	52.2 ± 10.3a	20.5	71.2 ± 7.1	19.8	
50–100	80.9 ± 38.0a	26.4	192.1 ± 24.2b	45.7	120.3 ± 31.9ab	39.5	64.0 ± 41a	25.1	145.3 ± 17.6	40.5	
0–100	306.1	100	420.0	100	304.1	100	254.2	100	358.7	100	
Note:

For each soil depth, data with different superscript letters are statistically different among different stand age classes. Data are mean and standard errors.

Carbon stocks in the forest ecosystem

The carbon stock in these forest ecosystems did not show significant variation with the age classes, ranging from 345.6 to 510.1 t ha−1 with an average of 449.4 t ha−1 (Table 7). Carbon stock in trees increased significantly with the increase in stand age whereas those in shrub and herb layers did not show a changing trend with stand age. Carbon stocks in fine roots and litter varied from 3.8 to 5.8 t ha−1 and from 1.5 to 11.7 t ha−1, accounting for 1–1.2% and 0.3–3.1% of the total ecosystem carbon stock, respectively. Carbon stock in deadwood increased from 0.6 to 8.6 t ha−1 with development of the forests with the proportion from 0.2% to 2.3%. Soil was the largest contributor to the ecosystem carbon stock although the contribution decreased from 88.6% to 68.1% with the development of forests. The contribution of biomass carbon stock increased from 11.4% to 31.9% across the range of stand age-class and was significantly related to stand age (p < 0.001) according to regression analysis (Fig. 2C).

Table 7 Carbon stock of each layer (t ha−1) in the ecosystem for different age classes.

Ecosystem component	Stand age classes	
I (<40 yr)	%	II (40–60 yr)	%	III (60–80 yr)	%	IV (>80 yr)	%	Average	%	
Tree	28.1 ± 5.2a	8.1	79.9 ± 8.4b	15.7	92.4 ± 14.2b	22.3	93.9 ± 8.8b	25.2	76.3 ± 6.4	17	
Shrub	0.9 ± 0.4a	0.3	0.14 ± 0.1a	0	1.3 ± 0.4a	0.3	0.8 ± 0.3a	0.2	0.58 ± 0.2	0.1	
Herb	0.5 ± 0.3a	0.2	1.8 ± 0.3b	0.4	0.5 ± 0.1ac	0.1	0.3 ± 0.1ac	0.1	1.1 ± 0.2	0.3	
Fine root	3.8 ± 0.9a	1.1	5.5 ± 0.7a	1.1	4.1 ± 2.7a	1.0	3.9 ± 0.9a	1.0	4.4 ± 0.4	1.0	
Deadwood	0.6 ± 0.3a	0.2	1.3 ± 0.7ab	0.3	4.6 ± 0.9bc	1.1	8.7 ± 1.5c	2.3	3.7 ± 1.1	0.8	
Litter	5.6 ± 1.9ab	1.6	1.5 ± 0.2b	0.3	7.3 ± 3.2a	1.8	11.7 ± 7.0a	3.1	4.5 ± 1.1	1.0	
Biomass total	39.5 ± 7.5a	11.4	90.2 ± 8.5b	17.7	110.2 ± 11.8b	26.6	119.1 ± 8.5b	31.9	90.7 ± 6.5	20.2	
Soil	306.1 ± 81.1a	88.6	420.0 ± 31.0a	82.3	304.1 ± 54.7a	73.4	254.2 ± 42.4a	68.1	358.7 ± 25.5	79.8	
Ecosystem	345.6 ± 74.8a	100	510.1 ± 34.2a	100	414.2 ± 52.6a	100	373.3 ± 45.9a	100	449.4 ± 26.0	100	
Note:

For each component, data with different superscript letters are statistically different among different stand age classes. Data are mean and standard errors.

Vegetation carbon accumulation potentials and rate

Carbon sequestration potential decreased with the development of the forests, from 76.8 t ha−1 in age class I to 5.3 t ha−1 in age class IV with an average of 29.4 t ha−1 (Fig. 3A). The carbon sequestration rate decreased from 2.47 t ha−1 year−1 in the growth stage of 37–56 years to 0.15 t ha−1 year−1 in the growth stage of 70–104 years (Fig. 3B).

Figure 3 Biomass carbon sequestration potential (A) and rate (B).

I, II, III and IV indicate the stand age interval: <40, 40–60, 60–80 and >80 years while I–II, II–III and III–IV indicate the growth stage of forest during 37–56, 56–70 and 70–104 years, respectively.

Discussion

Carbon allocation and dynamics in ecosystem components

The proportion of carbon stock in various components of tree layer decreased in an order of stems (54.3%) > branches (19.6%) > roots (15.3%) > needles (10.8%) (Table 3). Similar patterns were observed in Picea crassifolia forests in the Qilian Mountains (Wagner et al., 2015). The proportion of carbon stock in stem increased with stand age whereas the proportion in roots of the tree layer decreased, suggesting that carbon allocation to the roots reduced with the increase of stand age. Similar results were also observed in Pinus tabuliformis (Zhao et al., 2014), Pinus strobus (Peichl & Arain, 2006) and several other forest types in northeast China (Wang, Fang & Zhu, 2008). Such a trend can owed to the tree growth strategy that more resources was allocated to roots in early stages of growth in order to maximize water and nutrient assimilation that support survival (Helmisaari et al., 2002; Mund et al., 2002). With the development, productivity of tree foliage biomass no longer increases, resulting in decreased demand for nutrient and water supply from roots (Claus & George, 2005; Kurz, Beukema & Apps, 1996; Vanninen et al., 1996).

The amount of carbon stored in tree components and total tree biomass increased rapidly with stand age, which was consistent with the results reported by Chen, Liang & Wang (2016) and Guo & Ren (2014). The contribution of carbon stock in trees to total ecosystem increased significantly, from 8.1% to 25.2% across the entire age sequence (Table 7), indicating that the tree layer could accumulate carbon constantly throughout the forest development. The increase of carbon stock in trees got slowly when stand age exceeds 60 years. Taylor, Wang & Chen (2007) and Rothstein, Yermakov & Buell (2004) also reported that this increasing trend could be described as a sigmoidal pattern, i.e., young forests displayed rapid growth up to a certain age following which time their growth rate decreased gradually. This phenomenon could be ascribe to possible hydraulic and nutrient limitations (Ryan, Binkley & Fownes, 1997; Yuan & Chen, 2012), rise of mortality resulted from tree ageing (Luo & Chen, 2011), competition and disturbance (e.g., insects, wind and fungi) (Metslaid et al., 2007), and/or steeper declines in photosynthesis than in respiration (Tang et al., 2014). The increase of total ecosystem carbon stock across the entire age sequence was mainly contributed by the carbon stock in trees, which highlighted the importance of tree biomass carbon in forest ecosystem carbon estimates.

Understory vegetation (shrubs and herbs) play an important role in maintaining biodiversity and sequestering CO2 (Hou, Xi & Zhang, 2015), but their carbon stocks account only about 0.1% and 0.3% of the total ecosystem, respectively (Table 7). There are studies suggesting that absence of the calculation for understory could lead to underestimation of carbon sequestration capacity (Gao et al., 2014; Park, 2015). Previous studies found that carbon stock in the understory (shrubs and herbs) decreased with the increase in stand age (Cao et al., 2012; Taylor, Wang & Chen, 2007). In this study, however, the carbon stock in the understory did not show a clear changing trend with stand age. The distinction in species ecological characteristics, light and nutrients probably contributed to this difference (Abdallah & Chaieb, 2012; Arx, Dobbertin & Rebetez, 2012). Our results showed that the carbon stocks in shrub and herb layers were independent of stand age.

As a significant component affecting carbon transfer from vegetation biomass to soil, carbon stock in litter ranged from 1.5 to 11.7 t ha−1 with the average value of 4.5 t ha−1 in this study, which was comparable with the average value (2.8 t ha−1) of Picea crassifolia forests in the Qilian Mountains (Wagner et al., 2015). Similar climate and tree species should be responsible for the result. However, the litter carbon stock in this study was less than the average of major forest types in China (8.21 t ha−1) (Zhou, Yu & Zhao, 2000). This can be attributed to the influence of species, stand age and climate (Jiang et al., 2013; Takahashi et al., 2010). Carbon stock in litter layer showed different trends (increase or decrease) with stand age (Seedre et al., 2015; Zhao et al., 2014). Unlike previous results, carbon stock in litter layer showed no regular trend with stand age in this study. Generally, carbon stock in litter layer is determined by the net balance between litter fall input and decomposition output, thus any factors affecting the amount of input and decomposition rate would affect the carbon storage of litter.

In this study, carbon stocks in fine roots and deadwoods roughly accounted for 1.0% and 0.8% of total carbon stock in the whole ecosystem, respectively (Table 7). Exclusion from forest carbon stock estimation, which is commonly dealt with, may result in underestimation of total ecosystem carbon stock. Therefore, it is recommended that fine root and deadwood mass be calculated in such studies as natural forests. Carbon stock in deadwood increased with the increase in stand age. Similar trend was reported for Norway spruce ecosystem (Seedre et al., 2015), showing significant correlation between the dead biomass carbon and stand age. The high mortality of trees in old forests and relatively slow decomposition caused by low temperature should be responsible for the phenomenon in the areas. Fine root carbon stock in this study had no significant changing trend among age classes, which was inconsistent with that found in Betula platyphylla (Sun et al., 2015) and Scots pine (Makkonen & Helmisaari, 2001). This difference may be partly ascribed to the differences in species composition and/or environmental conditions (Leuschner et al., 2007).

The spruce forests have substantial soil organic carbon stock. The average amount in 0–100 cm is 358.7 t ha−1 in this study which is close to the results reported by previous relative studies. For example, Zhou, Yu & Zhao (2000) found that soil organic carbon stock in Picea–Abies forests in China was 360.8 t ha−1, and Wagner et al. (2015) found that soil organic carbon stock in Picea crassifolia forest in the Qilian Mountains was 305.0 t ha−1. Meanwhile, these results are greater than that reported for Norway spruce forest (130.0 t ha−1) (Seedre et al., 2015) and an averaged amount for major forest types in China (193.6 t ha−1) (Zhou, Yu & Zhao, 2000). The high soil organic carbon stock in spruce forests in northwest China is probably resulted from the low soil temperatures in combination with low soil moisture, which not only inhibited soil biological activity in large part of the growing season (Zheng et al., 2014), but also reduced the respiration rate and thus lowered the decomposition activity in soil (Rotenberg & Yakir, 2010; Wagner et al., 2015). The annually mean temperature and mean annual precipitation in our study area are all less than that in Norway spruce forest (Seedre et al., 2015). Callesen et al. (2003) and Perruchoud et al. (2000) also suggested that high soil organic carbon stock was often associated with low temperature and low soil moisture.

The average proportion of soil carbon stock in ecosystems of all stand age classes is 79.8% in this study (Table 7). This proportion is close to the results for Picea–Abies forests previously reported (77.8%) (Zhou, Yu & Zhao, 2000) and is higher than the global average value (approximately 69%) (Dixon et al., 1994). Meanwhile, this proportion is also much higher than the proportion for tropical forests (32%) reported by Pan et al. (2011), which was ascribed to the higher productivity of tree layer in tropical forests (Lü et al., 2010). The ratio of soil organic carbon stock to biomass carbon stock decreased from 8.7 to 2.4 with increasing stand age, which was consistent with boreal and temperate forests in northeastern China (Wei et al., 2013). The potential cause of the decline pattern might be the increasing trend of carbon stock in tree layer coupled with relatively stable carbon stock in soil layer along with the forest development.

Stand age is not a good predictor of soil organic carbon stock. Soil organic carbon stock did not change significantly with the increase of forest age in this study (Table 7), which was consistent with the conclusion by Marín-Spiotta, Sharma & Ramankutty (2013) found in successional and plantation forests. Lal (2004) and Martin et al. (2005) also proposed that soil organic carbon stock remains stable in place over years to centuries. One explanation was the formation of soil carbon needs a relatively long time, which is primarily controlled by three stabilization mechanisms: chemical stabilization, physical protection and biochemical stabilization (Six et al., 2002; Sollins, Homann & Caldwell, 1996; von Lutzow et al., 2006). This means that soil organic carbon stock may not be sensitive to the development of forests. Wang et al. (2016a) reported that soil organic carbon accumulation showed a time lag about 15–30 years compared to forest development in natural restoration. Additionally, many studies demonstrated there were other variation trends between stand age and soil organic carbon stock in forest ecosystem. For instance, Eaton & Lawrence (2009) found soil organic carbon stock was higher in the youngest and oldest natural secondary forests and was lower in sites of intermediate age in the southern Yucatán Peninsula. Nevertheless, the study of Wang et al. (2016a) found an initial decrease and subsequent increase trend of soil carbon stock with natural restoration chronosequence on the Loess Plateau of China. Zhou et al. (2006) found the soil carbon stock continued increasing even in an old-growth forest. Finally, studies also suggest that soil organic carbon stocks in forests can be influenced by climate, soil texture, soil thickness and tree species (Peichl & Arain, 2006; Seedre et al., 2015; Yang et al., 2008). Especially, Marín-Spiotta, Sharma & Ramankutty (2013) believed that climate could explain the greater variability in soil organic carbon stock in tropical forests than stand age. Overall, the mechanism behind the relationship between soil organic carbon accumulation and forest age needs further clarification.

The average value of ecosystem carbon stock in this study was 449.4 t ha−1 (Table 7), which was much more than other reports for natural secondary forest in northwest China (160 t ha−1) (Houghton & Hackler, 2003) and major forest types in China (258.8 t ha−1) (Zhou, Yu & Zhao, 2000). Meanwhile, our result was also higher than the reports for Picea crassifolia forest (348 t ha−1) (Wagner et al., 2015) and Norway spruce (393 t ha−1) (Seedre et al., 2015). Besides forest types and age structure, climate and edaphic factors may be responsible for such differences. Our study reconfirmed that spruce forests have substantial carbon stock.

Ecosystem carbon stock in this study had no significant variation with the increase of stand age (Table 7). The different variation pattern from Wang et al. (2016a) probably reflects the distinction of carbon sequestration ability of different tree species. The dominant tree species were Quercus liaotungensis, Populus davidiana and B. platyphylla in the study of Wang et al. (2016a). This may help us understand the spatial and temporal heterogeneity of carbon sequestration characteristics of different dominant tree species.

Carbon sequestration potentials in the forest biomass

Estimating the carbon sequestration potential in natural forest ecosystems is the basis of predicting long-term changes of carbon dynamics. The average value of carbon sequestration potential by the vegetation biomass was estimated as 29.4 t ha−1 in these natural spruce forests, which fell in the range (12.27–49.88 t ha−1) found in alpine forests in Qinghai-Xizang Plateau (Wang et al., 2016b). Meanwhile, these estimates were much less than the result reported by Zhong, Zhou & Li (2014) for natural forests in Karst area (57.88 t ha−1). The land-use history (Roxburgh et al., 2006), climate and age structure (Liu et al., 2014) are important factors affecting forest carbon sequestration potential in different study areas. Spruce forests in northwest China suffered some disturbances before implementing the NFP, resulting in lack of enough old stands. The low aged forests are the dominant contributor of carbon sequestration potential in the natural spruce forests in northwest China.

Carbon sequestration rate in these forests ranged from 0.15 to 2.47 t ha−1 year−1, which was less than the estimate of aboveground biomass for dominant tree species in a tropical deciduous forest (1.47–4.64 t ha−1 year−1) (Devi & Yadava, 2015). However, the maximum carbon sequestration rate (2.47 t ha−1 year−1) in this study was greater than the estimate of 0.95 t ha−1 year−1 for broad-leaved mature forests in Qinghai-Xizang Plateau (Wang et al., 2016b), and was also greater than that for temperate forests in northwest China (1.1 t ha−1 year−1) (Houghton & Hackler, 2003). These observed differences in the carbon sequestration rate may be related to species composition, growth stage and climatic factors (Devi & Yadava, 2015). Quantifying carbon sequestration rate across stand age gradient is expected to provide more accurate assessments of the potential increment of vegetation carbon accumulation in forests.

The spruce forests in this study were located in the areas of NFP, and the estimation of carbon sequestration potential and carbon sequestration rate was based on the premise of the forests being under the protection in the future. Forests under NFP have covered 45% of the total forest area in China and play a crucial role in regulating forest carbon sequestration (Chinese Ministry of Forestry, 2010). By 2020, carbon stock of forests under NFP is expected to increase by 4 × 109 t (Yang, 2017). The NFP has acted its positive role in carbon accumulation in forests as shown by the national forest inventory data (Hu & liu, 2006; Ke et al., 2015; Yang, 2017; Zhou et al., 2014). Our results highlighted the important role of NPF on carbon accumulation in forest ecosystems.

Conclusions

Carbon partitioning in all ecosystem components (vegetation biomass, deadwood, litter and soil) across age classes were comprehensively estimated in natural spruce forests in northwest China. Tree layer could accumulate carbon constantly with the development of forest. Shrub, herb, and litter layers accounted for about 0.1–1% of the total ecosystem carbon and were age-independent. Fine roots and deadwoods also play important roles in carbon accumulation, accounting for approximately 1.0% and 0.8% of ecosystem carbon stock, respectively. In each age class, the soil contributed more than 65% of the ecosystem carbon stock. Stand age is not a good predictor of soil carbon stock. Hence, accurate evaluation of the soil organic carbon stock dynamics in the forests requires long-term monitoring in situ. The carbon stock in biomass was the major contributor to the increment of carbon stock in ecosystems. The average value of carbon sequestration potential for these forests was estimated as 29.4 t ha−1, with the lower aged ones being the dominant contributor. The maximum carbon sequestration rate was 2.47 t ha−1 year−1 appearing in the growth stage of 37–56 years. The information from this study will improve our understanding of carbon stocks and dynamics in natural forests and can be helpful to evaluating the role of NFP in increasing forest carbon accumulation.

Supplemental Information

Supplemental Information 1 Raw data.

Click here for additional data file.

We gratefully acknowledge many other members (graduate and undergraduate students) from Northwest A&F University contributing to the field investigation.

Additional Information and Declarations

Competing Interests

Author Contributions

Data Availability

The authors declare that they have no competing interests.

Jun-Wei Yue performed the experiments, analyzed the data, prepared figures and/or tables, authored or reviewed drafts of the paper, approved the final draft.

Jin-Hong Guan performed the experiments, analyzed the data, approved the final draft.

Lei Deng performed the experiments, approved the final draft.

Jian-Guo Zhang performed the experiments, approved the final draft.

Guoqing Li performed the experiments, authored or reviewed drafts of the paper, approved the final draft.

Sheng Du conceived and designed the experiments, performed the experiments, authored or reviewed drafts of the paper, approved the final draft.

The following information was supplied regarding data availability:

The raw data is provided as a Supplemental File.

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
