# Peer review of "Allocation pattern and accumulation potential of carbon stock in natural spruce forests in northwest China"

_PeerJ, doi:10.7717/peerj.4859_

## Round 0.1 · original submission · Major Revisions

The determinants of forest carbon storage underpin forest management practices and thus are crucial in the context of climate change. While this manuscript addresses such determinants there are major questions regarding the geospatial relationship between sites and, most importantly, the sampling method employed for soil carbon analysis. As an example, reviewer three questions the soil particle size fraction used for analysis (see point 3 under Experimental Design). This point must be definitively addressed including perhaps reanalysis of the larger soil particle fraction. Please also carefully consider the points raised by all of the reviewers. Your revised manuscript will be most welcome considering the importance of the research to forest management practices.

Reviewer 1 ·

Basic reporting

no commen

Experimental design

no commen

Validity of the findings

no commen

Additional comments

This study has investigated the allocation pattern and accumulation potential of carbon stock in natural spruce forests in northwest China. The authors have comprehensively estimated the C partitioning in all ecosystem components (trees, shrubs, herbs deadwood, litter, and soil) across age classes in natural spruce forests in northwest China based on 39 plots. As we all know, the ecosystem components may play different role in the C stock and cycling of forest ecosystems. However, forest inventory is a hard work that needs to cost much time, labors, and also money. These results are helpful us to more accurately understand the carbon stock dynamics and sequestration potential in these natural forests. Overall, the methodology, data analyses, and scientific presentation are in a reasonable status. I recommend publication of this work after a minor revision.

1. The biomass equations of some other tree species (2. Populus davidiana, 3. Salix caprea and S. rehderiana, 4. Betula utilis) were also given in table 2; however, there is not any information in materials and methods part. So how are about the distribution proportion of these tree species in the Spruce forest?
2. In the manuscript, the soil coring technique was used to estimate fine root biomass. However, how the authors separated the the root of tree, shrub and herb when they estimated the fine biomass of each vegetation types?
3. In Line 175, Equation (2), the forest biomass C stock was calculated as the sum of biomass of tree component, shrub component, herb component, deadwood, fine root and litter. However, root biomass had been calculated in tree component, shrub component, and herb component. Thus, is there any double counting of root biomass in calculating the forest biomass C stock?
4. In the conclusion, the author concluded stand age is not a good predictor of soil C stock, and Fig. 2 also showed the obviously difference of SOC among different stand age classes. However, in a recent study, Wang et al (2016) reported the ecosystem carbon stock showed an initial decrease and then increase trend with the soil sampling depth of 0–100 cm in natural forest succession in Chines Loess Plateau. I suggest that the authors might discuss the present study with the previous works.

Wang et al. 2016. Dynamics of ecosystem carbon stocks during vegetation restoration on the Loess Plateau of China. Journal of arid land, 8(2), 207-220

Reviewer 2 ·

Basic reporting

Mixed of unprofessional sentence
A lot of grammar error.
Literature and reference were very old , > 5 years

Citation formatting of more than 2 citation for one sentence

Raw data and figures shared
White space after figure. Kindly check the upload instruction.
The author tends to write all the findings in the graph and data into words with minimal discussion. This style of writing although acceptable, it creates overlap explanation.
The acronym is, C for carbon, SOC and etc were used a lot and introduced incorrectly
Paragraph is not justifiable

Experimental design

Data analysis and review article type.
Only 2 types of trees were used. What are the % of the chosen tree types compared to the population?
Mean and standard error, do you have any indicator that which level is acceptable?

Validity of the findings

Some information in the article were assumption that if you collect this much of data it should represent the rest of the data.

The sample size determination should be explained

How does your findings make any different in terms of policy with the current understanding of carbon stock in China sequestration

Benefits of having your data versus the current High Carbon Stock (HCS) approach.

Additional comments

Difficult to read the article when too many acronym is used.

Biomas usually associated with energy. Stick with carbon stock.

Reviewer 3 ·

Basic reporting

The manuscript “Allocation pattern and accumulation potential of carbon stock in natural spruce forests in northwest China” by Yue et al. reports a comprehensive assessments of C storage in spruce forests characterized by stand age. The data used here were basically acquired as a part of large national project (NFP) that has been operating for last 20 years. Estimation and control of forest C stock is a very timely topic related to contemporary climate change and forest management issues. Therefore the research theme, presented in the article is well suited for PeerJ. However the paper has some basic weaknesses which must be overcome before considering for publication.
1. English language needs thorough revision. Some sentence structures are poor and not qualified as professional English. For example, Line 22-23, 51-53, 59, 63-65,268,320-321.
2. The arguments to justify the research as mentioned in Introduction (line 66-69 and 78-80) is not correct. All these components are generally included in stock assessment protocols (See IPCC Guidelines for National Greenhouse Gas Inventories).
3. The importance of fine root and dead wood can be moved to introduction section. Discussions should contain the critical analysis of the findings. From Line 283-291.
4. Most of the discussions consists of repetition of same things under two sections (4.2 Allocation pattern and 4.3 Stock dynamics), my suggestion is try to combine the two sections and precisely discuss the each components.
5. Figure 1 please explain what does the 15 dot points in sketch map indicate .
6. Figure 2 Please Split it or convert to Table.
7. Figure 3 Caption of the figure should be self-explanatory. Please mention what is CSP, CSR, age classes (i, ii.... & i-ii, ii-iii etc.), add error bar and statistics.
8. Table 1 H (m) should be Height (m), unit of basal area cm2?
9. Table 2 please mention only those equations that have been used in the current study.
10. Table 4 Table caption should be More detailed (explain age class and statistics).

Experimental design

1. The location of the study areas are not completely described. Location map showed that all 39 sampling plots were spread over large area of 4 locations, nearly 300-700 km apart from each other. It’s not clear how the 4 plots of 20mX50m (age class IV, for example) were distributed over those 4 locations and how these are comparable if not in same geographic location.
2. No information about systemetic sampling/selecting plots, I mean the protocol for selection of 20 mX 50 m plot in a large forest area. For example in a similar study (Cui et al PLoS ONE 10(9): e0137452) in Shaanxi province, 28.28 m×28.28 m plots were established in each 4km×8 km grid along a transect line.
3. By the universal definition, all particles less than 2mm are generally considered as soil (fine earth). So sieving the air dried soil samples by a 0.25mm sieve (Line 166) before chemical analysis is clearly a wrong procedure with the possibilities of an erroneous estimation. Because small aggregates, particulate organic matter can be lost due to use of fine screen. (Please see the sieve size in Global Change Biology (2014) 20, 2644–2662, CATENA (2016) 137: 651-659).
4. The Decomposition class and specific density of dead trees need detailed procedures as the referred literature in not available in English (Line 133-136).
5. No sampling time (year, month) for soil or root sampling was mentioned.
6. Please mention how fine roots were sorted (hand picking/wet or dry sieving)(Line 156).
7. The method of estimating C sequestration potential (CSP) using the equation-3 needs reference (Line 186).

Validity of the findings

Minor revisions.
1. Not clear (Line144)
2. Not clear (Line 206)
3. No regression analysis was performed (in section 2.5 Statistical analysis) (Line 215).
4. At each soil depth? (Line 219)
5. No Regression analysis ? (Line 239)
6. Line 331 (In this study-----) and Line 333 (Such differences---) is contradictory.
7. Line 336 -338 not clear.
8. Line 359 Please explain how soil formation correlate with forest development in the present study sites. Need reference.
9. L397 Age class iv is less than 70%.

Additional comments

The authors investigated ecosystem C accumulation pattern and potential in spruce forests in China. The subject-matter is interesting and fit with the scope of PeerJ. The Authors correctly identified and addressed the major components of ecosystem C stock. But as the study location covered a large geographical area, a systematic survey-protocol should be followed during the plot selection and/or sampling, with the description of forest, landscape, soil and climate of each sampling sites. In addition, methodological robustness in sampling-preparing-analyzing is crucial, where I found a lacking in soil sample processing (pass through 0.25mm screen instead of standard 2mm sieve) which might have influence on objectives of the study.

---

## Round 0.2 · accepted · Accept

Congratulations! Your manuscript has been accepted for publication in PeerJ. Note, however, minor revisions suggested by Reviewer 3 need to be addressed while in production.

# Reviewer 1 ·

Basic reporting

The authors responsed all the commentary carefully in the revised manuscript, i recommend the manuscript to be accepted after this revision.

Experimental design

The authors responsed all the commentary carefully in the revised manuscript, i recommend the manuscript to be accepted after this revision.

Validity of the findings

The authors responsed all the commentary carefully in the revised manuscript, i recommend the manuscript to be accepted after this revision.

Additional comments

The authors responsed all the commentary carefully in the revised manuscript, i recommend the manuscript to be accepted after this revision.

Reviewer 2 ·

Basic reporting

Better writing style then previous version

Format in 455 : reference
paragraphs were missalign

Experimental design

Within the journal scope.
Author had provided explanation in rebuttal

Validity of the findings

The result is valid

Additional comments

The impact of the study may justify the government program, but thats all.

Reviewer 3 ·

Basic reporting

Authors' actions are satisfactory

Experimental design

Authors' actions are satisfactory

Validity of the findings

Authors' actions are satisfactory

Additional comments

The manuscript has been improved satisfactorily. Just three minor comments:
1. In line 197 (pdf version) ……in the soil layer m (totally five sampling layers in this study)- not totally- should be total five….
2. In different Tables ‘Stand age classes’ were presented as I, II, III etc. which were explained only in Table 1. I think you can use either I (<40 yr), II (40-60 yr)….. this style in all Tables or mention this in Table captions.
3. In Table 6 footnote please pest ‘Data with different lowercase ……………..’